# Testing the Interface Hypothesis: Evidence from processing directions of possession transfer in double object constructions by L1-Mandarin Chinese L2-English learners

Yuxi Li, Tao Zeng *, Ze Liu

Department of Foreign Languages, Hunan University, Changsha, Hunan Province, The People's Republic of China

* taozengclarry@hnu.edu.cn

**Data Availability Statement:** All relevant data are within the paper and its Supporting Information files.

## Abstract

The Interface Hypothesis postulates that internal interfaces linking domains within the language system are ultimately acquired at a near-native proficiency level in second language (L2) acquisition. While extensive research in the field of L2 acquisition has demonstrated that L2 learners often fail to completely acquire internal interfaces, the present study adds to this line of research by delving into an understudied phenomenon at the syntax-semantics interface, namely, directions of possession transfer in English Double Object Constructions (DOCs). Specifically, this study focuses on the processing of English DOCs containing verbs with varying or opposing directions of possession transfer. Employing a self-paced reading task and a comprehension task, we examined the performance of 30 native English speakers and 63 first language (L1) Mandarin Chinese learners of English. The findings suggest that L2 learners struggle to fully acquire the intricacies of English DOCs. Additionally, this study reveals that learners encounter challenges in processing English DOCs with left-directional verbs such as *buy*, compared to those with right-directional verbs such as *post* and non-directional verbs such as *build*. This points to an indeterminacy in the processing of the syntax-semantics interface constructions by L2 learners at developmental stages. To account for these findings, we propose an interface indeterminacy account, contending that the indeterminacy in the processing of the syntax-semantics interface constructions by L2 learners stems from the inherent indeterminacy in these constructions. Furthermore, this indeterminacy is likely to be vulnerable to a number of variables, including L1 transfer, learners' L2 proficiency, and processing resources.

## 1 Introduction

Recent trends in second language (L2) acquisition research has witnessed a surge in investigations into interfaces, which marks a pivotal frontier in exploring the factors contributing to non-native-like performance among L2 learners. However, a consensus among researchers

**Funding:** This work was supported by the Humanities and Social Science Fund of the Ministry of Education of China 24YJC740042 awarded to YL. No additional external funding was received for this study. The funder did not play a role in the study design, data collection and analysis, decision to publish, or preparation of the manuscript.

**Competing interests:** The authors have declared that no competing interests exist.

remains elusive regarding the extent to which L2 learners are non-native-like in the acquisition of interfaces. Within the Interface Hypothesis (IH), some studies contend that interfaces linking syntax and other cognitive systems are less likely to be acquired completely in L2 acquisition compared to narrow syntax [1, 2]. Building upon this perspective, some researchers go further to propose that internal interfaces that can link domains within the language system (e.g., the syntax-semantics interface) cause fewer challenges at advanced stages of L2 development than external interfaces that can link linguistic domains with other aspects of world knowledge and cognition (e.g., the syntax-discourse interface) [3, 4]. While a growing body of evidence supports that internal interfaces can be acquired at a near-native level in L2 development [5, 6], conflicting findings imply that certain internal interfaces such as the syntax-semantics interface may not be completely acquired by L2 learners [7, 8]. However, the issue how L2 learners process phenomena within interfaces remains a relatively underexplored area in the existing literature.

The current study seeks to address this scholarly gap by delving into the processing of English Double Object Constructions (DOCs) within the syntax-semantics interface, specifically focusing on the directions of possession transfer in the grammars of first language (L1) Mandarin Chinese learners of English. Are the interface properties of the directions of possession transfer in English DOCs acquirable? If problems occur, what will pose the problems?

Directions of possession transfer in DOCs differ in English and Mandarin Chinese, as shown in (1) and (2). The Chinese examples in (1) indicate that transfer of possession relation between the direct object (DO) and the indirect object (IO) is directional, which is contingent upon the thematic role of the IO. For example, in (1a), the thematic role of the IO *Mary* is a Recipient, and the direction of possession transfer is rightward. In the contrary, in (1b), the thematic role of the IO *Mary* is a Source, and the direction of possession transfer becomes leftward. This contrasts with English DOCs, in which the directions of possession transfer are uniformly rightward, as in (2), irrespective of the thematic roles of the IO. Furthermore, a distinction between English and Chinese DOCs pertains to the verbs used. For instance, verbs associated with creation, such as 'knit' (*zhi* in Mandarin Chinese), are compatible with English DOCs but incompatible with Chinese DOCs, as illustrated in (1c) and (2c). This distinction highlights that certain verbs, despite the applicability in English constructions, do not align with the structural requirements of Chinese DOCs.

1. Chinese DOCs

    a. John ji-le Mary yi-ben shu.
       John post-PERF Mary one-CLA book
       'John posted Mary a book.'

    b. John tou-le Mary yi-liang che.
       John steal-PERF Mary one-CLA car
       'John stole a car from Mary.'

    c. *John zhi-le Mary yi-jian waitao.
       John knit-PERF Mary one-CLA coat
       'John knitted Mary a coat.'

2. English DOCs

    a. John posted Mary a book.

    b. John stole Mary a car.

    c. John knitted Mary a coat.

Previous studies [9–11] indicated that L2 learners could not completely acquire English DOCs, which was attributed to the transferred L1 grammar. Building on this foundation, this study posits that the delayed acquisition of English DOCs is ascribed to the inherent indeterminacy within the syntax-semantics interface constructions. As the forthcoming analyses will demonstrate, English DOCs cause different degrees of difficulties for Chinese learners of English at developmental stages, which implies a pronounced degree of indeterminacy in the learners' processing of English DOCs. This proposition shifts the focus from an interface account to a more nuanced exploration of the interface indeterminacy, which provides a deeper understanding of the challenges encountered by L2 learners in the acquisition of English DOCs.

## 2 Literature review

### 2.1 Directions of possession transfer in English DOCs

English DOCs are widely recognized as syntactic constructions that consist of a ditransitive verb and three arguments. The verb typically denotes an event of possessive transfer (e.g., *offer*) or cognitive transfer (e.g., *tell*), with each argument assigned a distinct thematic role—Agent, Recipient, and Theme [12, 13]. The canonical structure of English DOCs is 'Subject + Verb + Indirect Object (IO) + Direct Object (DO)'. English DOCs can alternate with prepositional dative constructions (PDCs), which follow the structure 'Subject + Verb + DO + Preposition + IO'. Examples of the two constructions are provided in (3).

(3) a. John told Mary a story.

 b. John told a story to Mary.

The examination of English DOCs begins with an exploration of their syntactic and semantic properties. Syntactically, English DOCs exhibit a c-command asymmetry: the IO asymmetrically c-commands the DO, as in (4)-(6) [14, 15].

(4) a. I showed Mary herself. (anaphor binding)

 b. *I showed herself Mary.

(5) a. I showed each man the other's socks. (the *each. . .the other* construction)

 b. *I showed the other's friend each man.

(6) a. I showed no one anything. (negative polarity items)

 b. *I showed anyone nothing.

Semantically, English DOCs encapsulate a directional possession relation between the IO and the DO, a feature extensively discussed by researchers [15–18]. Specifically, the IO bears the role of an intended Recipient of the DO, signifying a purposeful transfer of possession. Illustratively, in (7), the thematic structure reflects the intended transfer of the Theme *a cake* from the Agent *John* to the Recipient *Mary*.

(7) a. John baked Mary a cake.

 b. John bought Mary a cake.

English DOCs impose specific semantic constraints, particularly concerning the thematic role of the non-core IO. This linguistic phenomenon restricts the thematic role of the IO to that of a Recipient. To elucidate, consider the verb *sell*, which inherently means 'give or hand

over (something) in exchange for money', as demonstrated in (8a). In the context of English DOCs, such as in (8b), the sentence implies that 'John gives or hands over a house to Mary in exchange for money', wherein the IO *Mary* undertakes the thematic role of a Recipient. Verbs like *sell*, which convey a sense of 'giving', are termed rightward verbs, denoting a transfer direction of the Theme from the Agent to the Recipient. Conversely, consider the verb *buy*, with a meaning of 'John obtained a skirt by paying money for it' in (9a). Despite being an action of 'obtaining', as in (9b), the sentence expresses that 'the skirt was transferred from John to Mary'. Such verbs, categorized as leftward verbs, still result in rightward directions of possession transfer when used in English DOCs. Furthermore, non-directional verbs, such as *cook*, which denotes 'preparing a meal', as in (10a), also conform to the rightward directionality when used in English DOCs. As in (10b), the sentence conveys that 'John is an intended participant eating a meal cooked by Mary'. Therefore, English DOCs exclusively allow right-directional constructions, irrespective of the inherent directionality of the verbs. This semantic intricacy highlights a consistent pattern in English, which emphasizes the directional flow of possession transfer in English DOCs, regardless of the inherent directional characteristics of the verbs involved.

(8) a. John sold a house.

 b. John sold Mary a house.

(9) a. John bought a skirt.

 b. John bought Mary a skirt.

(10) a. John cooked a meal.

 b. John cooked Mary a meal.

The IO of DOCs is derived via Appl(icative) projection [15–18]. In this theoretical framework, the IO emerges as a product of the Applicative projection, represented by a functional head Appl introduced under a transitive verb. Unlike core arguments that is an integral part of the verb's argument structure, the Appl head introduces a non-core argument, the IO. This is particularly evident in English transitive verbs such as *write*, typically involving two core arguments: Agent and Theme. For instance, in the sentence "John wrote a letter", both *John* and *the letter* serve as core arguments essential to the verb's basic meaning. However, this argument structure encounters a challenge when extended to sentences like *John wrote Mary a letter*, in which *Mary* appears less integral to the fundamental meaning of the verb. High applicatives are distinguished from low applicatives [18]. High applicatives convey a thematic relation between an individual and the event, whereas low applicatives imply a transfer of possession relation between the DO and the IO. This difference in Appl heads provides a valuable diagnostic tool, which allows for the differentiation of DOCs in English and Mandarin Chinese. The semantics pertaining to high and low applicatives offer insights into the nature of these constructions, contributing to our understanding of the syntactic and semantic nuances that underlie the structure of DOCs.

There is a diagnostic approach based on the (in)compatibility of applicative heads with unergative and stative verbs, offering valuable insights into distinguishing between high and low applicatives [18]. According to this diagnostic, the low applicative head, implying a possession relation between the DO and the IO, exhibits incompatibility with unergative verbs. In contrast, the high applicative head that is associated with a thematic relation between an individual and the event demonstrates compatibility with unergatives. Morever, the low applicative head makes no sense with verbs that are static, because the type of event denoted by a static

predicate is inconsistent with the theme undergoing change of possession. In contrast, the high applicative head can combine with stative verbs.

In accordance with the classification of high and low applicatives, English DOCs exemplify the characteristics of low applicatives, which denote a transfer of possession relation between the IO and the DO. The low applicative nature of English DOCs is evidenced by the fact that they are not incompatible with unergatives and stative verbs, as demonstrated in (11).

(11) a. *John ran Mary. (incompatibility with unergatives)

 a. b. *John held Mary a bag. (incompatibility with stative verbs)

The current analysis posits that English DOCs align with the characteristics of a specific type of low applicative. In this configuration, the IO is an intended Recipient of the DO, and the directions of possession transfer are consistently rightward. It is noteworthy that English DOCs represent a singular manifestation within the broader category of low applicatives. Another subtype of low applicative involves a different semantic relationship, wherein the IO assumes a Source rather than a Recipient relation to the DO [18]. This nuanced distinction corresponds to two subtypes of low applicative heads, denoted as LowAppl-TO and LowAppl-FROM [19]. The bifurcation within the low applicatives highlights the versatility of this structure in denoting different thematic relations between the IO and the DO. The specific manifestation of English DOCs as LowAppl-TO constructions emphasizes the rightward direction of possession transfer, with the IO consistently portraying an intended Recipient. This nuanced classification enhances the understanding of the intricacies in the low applicatives, and provides a comprehensive perspective on the syntactic and semantic dimensions of English DOCs.

This article adopts the high and low applicative classification as a foundational framework, providing a solid ground to systematically compare the semantic properties of English and Chinese DOCs. This principled approach enables an examination of the syntactic and semantic nuances inherent in the two languages, which contributes to a comprehensive understanding of their respective applicative structures.

In addition to the applicative analysis, it is imperative to differentiate DOCs from Ditransitive Constructions [20]. In the case of DOCs, verbs such as 'bake' initially take two arguments, as exemplified in (12b). The applicative head introduces a non-core argument based on these two original arguments, giving rise to English DOCs as in (12a). In contrast, Ditransitive Constructions involve verbs like 'give', which inherently take three arguments, as shown in (13a). Notably, sentences lacking the IO, as shown in (13b), are deemed unacceptable in Ditransitive Constructions. Whereas these two constructions may exhibit surface similarities, a crucial distinction arises in the nature of the arguments. In DOCs, the Indirect Object (IO) is a non-core argument projected by the applicative head, stemming from the original two arguments of the verb. On the contrary, Ditransitive Constructions involve two core arguments. These demonstrates the inherent difference in the syntactic as well as semantic structures between DOCs and Ditransitive Constructions.

(12) a. John baked Mary a cake.

 b. John baked a cake.

(13) a. John gave Mary a cake.

 b. *John gave a cake.

In summary, English DOCs align with the characteristics of low applicatives, in which the IO is an intended recipient of the DO. The consistent rightward direction of possession

transfer in English DOCs underscores their classification as low applicatives. This succinctly captures the essential semantic and syntactic features that distinguish English DOCs within the applicative framework.

## 2.2 Directions of possession transfer in Chinese DOCs

Chinese DOCs and English DOCs, at a surface level, exhibit similarities, featuring a ditransitive verb and three arguments. In spite of these apparent parallels, a significant distinction emerges in their applicative structures. English DOCs primarily manifest as low applicatives, establishing a connection between the additional individual (i.e., the IO) and the DO, and the DO is intended to be possessed by the new individual (i.e., the IO). In contrast, Chinese DOCs encompass both low and high applicatives. Specifically, Chinese DOCs encompass not only LowAppl-TO but also LowAppl-FROM. As shown in (14a) and (14b), the IO *Mary* is a Recipient and a Source, respectively. In the case of high applicatives, the high applicative head relates new event participants to the event described by the verb. As illustrated in (14c) and (14d), the IO *Mary* is a Benefactive and a Malefactive, respectively. This nuanced classification captures the versatility of Chinese DOCs, encompassing a spectrum of thematic relations and possession transfers in both low and high applicative structures. The comparison of Chinese and English DOCs within the applicative framework provides valuable insights into the syntactic and semantic variations that underlie these constructions.

(14) a. John fu-le Mary yi-bi qian.
 John pay-PERF Mary one-CLA money
 'John paid Mary a sum of money.'

b. John qiang-le Mary yi-bi qian.
 John grab-PERF Mary one-CLA money
 'John grabbed a sum of money from Mary.'

c. John shang-le Mary yi-ge ping-guo.
 John award-PERF Mary one-CLA apple
 'John awarded Mary an apple.'

d. John chi-le Mary yi-ge ping-guo.
 John eat-PERF Mary one-CLA apple
 'John ate an apple from Mary.'

As observed, the IO in Chinese DOCs exhibits a remarkable flexibility, assuming thematic roles as both a Recipient and a Source. This flexibility allows Chinese DOCs to accommodate both leftward and rightward directions of possession transfer. In (15a), the theme, represented by *a book*, is conveyed from the left argument (Agent *John*) to the right argument (Recipient *Mary*). Conversely, in (15b), the theme *a book* undergoes possession transfer from the right argument (Source *Mary*) to the left argument (Agent *John*). Moreover, the IO *Mary* can interchangeably bear the role of a Recipient or a Source, as evident in (15c). This sentence can be interpreted either as 'John borrowed a book from Mary' or as 'John lent Mary a book'. This nuanced flexibility adds depth to the range of meanings expressed by Chinese DOCs. Concerning the directions of possession transfer, verbs in Chinese DOCs can be categorized into leftward verbs (e.g., "偷" (tōu) 'steal'), rightward verbs (e.g., "寄" (jì) 'post'), and left-rightward verbs (e.g., "借" (jiè) 'borrow' or 'lend'). The rightward direction signifies the concept of 'giving', whereas the leftward direction conveys the meaning of 'obtaining'. This classification provides a perspective for understanding the diverse semantic nuances embedded in the directions of possession transfer within Chinese DOCs.

(15) a. John ji-le Mary yi-ben shu.
       John post-PERF Mary one-CLA book
       'John posted Mary a book.'

b. John tou-le Mary yi-ben shu.
       John stole-PERF Mary one-CLA book
       'John stole a book from Mary.'

c. John jie-le Mary yi-ben shu.
       John jie-PERF Mary one-CLA book
       'John borrowed a book from Mary or John lent Mary a book.'

As evident in the examples above, a distinctive feature emerges in the directional possibilities of DOCs in English and Chinese. While English exclusively includes right-directional DOCs, Chinese includes both right-directional and left-directional DOCs. In Chinese DOCs, the flexibility is illustrated in both directions of possession transfer. Example (16a) exemplifies the rightward direction, in which the theme, represented by "蛋糕" (dàngāo or cake), is transferred from the left argument (Agent *John*) to the right argument (Recipient *Mary*). Conversely, in (16b), the leftward direction is portrayed as the theme "蛋糕" undergoes possession transfer from the right argument (Source *Mary*) to the left argument (Agent *John*). In contrast, English DOCs are constrained to right-directional constructions, as demonstrated in examples (17a) and (17b). Both instances depict the rightward direction, where the DO (i.e., *a cake*) is solely transferred from the right argument (John) to the left argument (Mary). This contrast highlights the unique semantic capabilities of Chinese DOCs, which encompass both obtaining and giving scenarios, in contrast to the exclusively giving nature of English DOCs. The nuanced directional possibilities contribute to the intricacies of possession transfer within these constructions in the two languages.

(16) Chinese DOCs

    a. John ji-le Mary yi-he dangao.
       John post-PERF Mary one-CLA cake
       'John posted Mary a cake.'

    b. John mai-le Mary yi-ge dangao.
       John buy-PERF Mary one-CLA cake
       'John bought a cake from Mary.'

(17) English DOCs

    a. John posted Mary a cake.

    b. John bought Mary a cake.

Another notable distinction arises in the verb compatibility. The verbs compatible with English and Chinese DOCs are not interchangeable; certain verbs can be used with English DOCs but find no compatibility with Chinese DOCs, and vice versa. In English DOCs, non-directional verbs, particularly those denoting creation actions (e.g., *bake*, *build*), seamlessly integrate, as demonstrated in (18a-b) and (19a-b). Conversely, these non-directional verbs of creation, when applied to Chinese DOCs, face incompatibility, as illustrated in (19a-b). On the contrary, verbs of consumption (e.g., "吃" (chī) 'eat', "喝" (hē) 'drink', "用" (yòng) 'use') find compatibility within Chinese DOCs but are incompatible with English DOCs, as depicted in (18c-e) and (19c-e). This disparity in verb compatibility sheds light on the distinct semantic

constraints that govern English and Chinese DOCs, further contributing to the intricate variations between the two constructions.

(18) English DOCs

 a. John baked Mary a cake.

 b. John built Mary a house.

 c. *John ate Mary an apple.

 d. *John drank Mary a bottle of water.

 e. *John used Mary a notebook.

(19) Chinese DOCs

 a. *John kao-le Mary yi-kuai dangao.
 John bake- PERF Mary one-CLA cake.
 'John baked Mary a cake.'

 b. *John jian-le Mary yi-zuo fangzi.
 John jian- PERF Mary one-CLA house
 'John built Mary a house.'

 c. John chi-le Mary yi-ge pingguo.
 John eat-PERF Mary one-CLA apple.
 'John ate an apple from Mary.'

 d. John he-le Mary yi-ping shui.
 John eat-PERF Mary one-CLA water.
 'John drank a bottle of water from Mary.'

 e. John yong-le Mary yi-ge benzi.
 John use-PERF Mary one-CLA notebook.
 'John used a notebook from Mary.'

In summary, DOCs in English and Mandarin Chinese showcase distinct syntactic and semantic properties. English DOCs adhere to the classification of low applicatives, whereas Chinese DOCs demonstrate a broader applicative spectrum, incorporating both low and high applicatives. This versatility is shown in the IO of Chinese DOCs, which can function as either a Recipient or a Source. Additionally, the directions of possession transfer diverge between the two languages, with English DOCs consistently following a rightward trajectory, while Chinese DOCs exhibit the flexibility to be either rightward or leftward in this aspect. These nuanced differences can contribute to a comprehensive understanding of the syntactic and semantic variations inherent in DOCs across English and Mandarin Chinese.

## 2.3 Summary: Directions of possession transfer in English and Mandarin Chinese

The distinctive properties of English and Chinese DOCs are succinctly presented in Table 1. While English DOCs consistently exhibit rightward directions of possession transfer, independent of the original tendencies of the verbs, Chinese DOCs display an evident versatility, licensing both rightward and leftward directions. Additionally, the contrast extends to non-directional verbs, which find a place in English DOCs but are precluded in their Chinese counterparts. For instance, regarding rightward verbs such as *post*, both English and Chinese DOCs

**Table 1. Directions of possession transfer in English and Mandarin Chinese.**

| Verb type | English | Mandarin Chinese |
|---|---|---|
| Rightward | Rightward | Rightward |
| Leftward | Rightward | Leftward |
| Non-directional | Rightward | / |

exhibit rightward directions. In contrast, the inclusion of leftward verbs such as *buy* maintains leftward directions in Chinese DOCs, while causing a shift to rightward directions in English DOCs. Further, non-directional verbs such as those denoting creation (e.g., *build*) are compatible with English DOCs but find no correspondence in Chinese DOCs.

To summarize, the intricate relationship between verbs and constructions exhibits language-specific variations. Notably, the distinctions between English and Mandarin Chinese become evident, as English DOCs consistently manifest rightward directions of possession transfer, irrespective of the original verb directions. In contrast, Chinese DOCs adhere to the original direction of verbs, which accommadate both rightward and leftward directions. The differences lay the foundation for making predictions regarding the L2 acquisition of English DOCs by Chinese speakers.

## 2.4 L2 acquisition of double object constructions

While there have been many L2-acquisition studies on the comparison between DOCs and PDCs [21–29], there have been relatively fewer studies administered to investigate the L2 acquisition of DOCs at the syntax-semantics interface [9, 10, 30].

Inagaki conducted a study on the L2 acquisition of English DOCs by L1-Chinese and L1-Japanese English learners [9], specifically concentrating on four narrow-range verb classes, namely the Throw class, the Tell class, the Whisper class and the Push class. Verbs in the Throw class can appear in English and Japanese DOCs but are incompatible with Chinese DOCs, while verbs in the Whisper class are exclusive to Japanese DOCs. Tell-class verbs are compatible with DOCs in all three languages, while Push-class verbs are incompatible. The results from an acceptability judgment task revealed that Japanese and Chinese speakers could distinguish DOCs containing Tell-class verbs from those with Whisper-class verbs, but struggled to distinguish DOCs with Throw-class verbs from those with Push-class verbs. The results suggested that the acquisition of English DOCs is affected by the properties of equivalent structures in the learners' L1. Importantly, L1 influence was more pronounced in the L2 Chinese grammars than in L2 Japanese grammars, given the distinctions in verb classes between the two languages.

Al-Jadani examined the L2 acquisition of English DOCs by Arabic speakers and the L2 acquisition of Arabic DOCs by English speakers [10]. In English, verbs in the Give class, Tell class, and Throw class can appear in both DOCs and PDCs. However, Arabic DOCs only allow certain verbs in the Give class and the Tell class. In addition, English prohibits Scrambling Dative Constructions, a structure permitted in Arabic. The results of the L2 English study indicated that L1-English L2-Arabic learners struggled to acquire structures absent in their L1, and generally did not learn what are disallowed in English. Conversely, the L2 Arabic study revealed that L1-English learners could not consistently identify the ungrammaticality of certain Arabic structures but were able to acquire SDCs. The findings suggested that L2 learners initially transfer the L1 grammar, gradually undergo restructuring, and ultimately arrive at the L2 grammar once effective positive evidence is provided.

Huang and Yuan administered a study investigating the interpretations of Chinese applicative DOCs by English and Spanish speakers [20]. The investigation employed an acceptability

judgment task and an animation matching task, involving participants at different proficiency levels: post-beginner, intermediate, and advanced. The findings suggested that L2 learners demonstrated their ability to adjust their L2 grammars to incorporate new target properties. Nevertheless, they had difficulty removing thematic relations transferred from their L1, indicating a potential permanence in the deviation of adult L2 grammars from those of native speakers. The difficulties encountered by L2 learners were accounted for via the Dormant-Feature Hypothesis [31]. This hypothesis posits that when the input fails to provide evidence either confirming or disconfirming a transferred property, that property loses its vigor and becomes dormant.

In light of the observed influence of L1 on the L2 acquisition of English DOCs, Oh conducted a study to explore the recovery process from the negative effects of L1 transfer [30]. The investigation focused on investigating Korean speakers' knowledge of semantic properties in association with English DOCs. Oh employed an acceptability judgment task for comparing the performance of Korean speakers and native English speakers. The results revealed that advanced L2 learners exhibited the ability to acquire semantic properties relevant to English DOCs, suggesting that the detrimental impact of L1 transfer can be mitigated with increased proficiency. This study thus implies that the acquisition of the semantics of a construction contributes to the mastery of its syntax.

Indeed, the acquisition of DOCs in L2 acquisition involves multiple factors, with L1 transfer being a significant but not exclusive contributor. The learners' proficiency also plays an important role and interacts with L1 transfer. Wolfe-Quintero conducted production tasks, revealing L1 influence on learners' developmental gains in producing target lexical structures and distinguishing between verbs [32]. Moreover, Kim et al. highlighted the impact of L2 proficiency on the integration of verbal and constructional information in the processing of English DOCs by Korean learners of English [11]. This collective evidence underscores the multifaceted nature of L2 acquisition, where both L1 transfer and L2 proficiency contribute to the L2 learners' ability to master complex linguistic structures like DOCs.

Prior studies in the acquisition of English DOCs primarily utilizes offline methods, with only a limited number of studies using online methods. Agirre investigated the acquisition of English DOCs by Spanish speakers across elementary, intermediate, and advanced proficiency levels via an auto-paced reading task and a self-paced reading task [33]. It was found that learners with the lowest proficiency demonstrated transfer effects from their L1. Kim et al. examined the influence of construction types on the integration of verbs and constructions in the processing of English DOCs and PDCs by L1-Korean L2-English learners [11]. They manipulated the verb's association strength within these constructions using a self-paced reading task. Results showed that Korean speakers spent significantly more time on the postverbal complements in English DOCs compared to PDCs.

To sum up, the studies discussed reveal varying factors influencing the acquisition of English DOCs, shedding light on different dimensions of L2 acquisition. However, a consensus is yet to be reached on the acquirability of the syntax-semantics interface, and the factors contributing to the processing of the syntax-semantics interface remain a subject of ongoing investigation.

## 3 This study

The current study aims to investigate the processing of directions of possession transfer in English DOCs involving the syntax-semantics interface by Chinese speakers. It is postulated that the syntax-semantics interface is unproblematic and is eventually acquired at near-native level in L2 acquisition [34, 35]. The current investigation, on the other hand, involves a

comparative analysis of outcomes obtained from native English speakers and Chinese learners of English at various proficiency levels, spanning from beginner to advanced stages. The central purpose is to investigate whether learners at developmental stages encounter challenges within the syntax-semantics interface, and whether their representations are in line with native-like patterns.

### 3.1 Research questions

Three exploratory research questions are addressed the current study:

1. Are there differences between native speakers and L2 learners in the processing of directions of possession transfer in English DOCs?

2. What factors influence the processing of directions of possession transfer in English DOCs? Specifically, will verb type, L1 transfer, and L2 proficiency influence Chinese speakers' processing of directions of possession transfer in English DOCs?

3. To what extent can Chinese learners of English acquire directions of possession transfer in English DOCs at the syntax-semantics interface?

Research question 1 aimed to compare the processing of directions of possession transfer in English DOCs between L2 and native speakers. Given that participants in the present study are at developmental stages, two potential outcomes were considered: either L2 and native speakers would display different patterns, or they would exhibit similar patterns, despite L2 learners not fully acquiring English DOCs.

Research question 2 examined main factors affecting the processing of directions of possession transfer in English DOCs, which include verb type, L1-L2 discrepancies, and learners' L2 proficiency. Previous studies have shown that L1 transfer impacts the L2 acquisition of DOCs in English. Participants' L2 English and L1 Mandarin Chinese are similar in the rightward direction but different in the leftward direction. As such, it was expected that English DOCs with rightward verbs would be processed better than with leftward verbs. In addition, prior studies have indicated the effect of learners' L2 proficiency. As such, it would also influence the processing of directions of possession transfer in English DOCs. Moreover, prior studies have found that L2 learners behaved differently with regard to English DOCs containing verbs in different sub-classes. As such, it was predicted that verb type would influence the processing of English DOCs containing verbs with different directions of possession transfers.

Addressing research question 3, the Interface Hypothesis postulates that internal interfaces such as the syntax-semantics interface are acquirable. Nonetheless, previous research has shown that L2 learners could acquire English DOCs containing verbs in certain sub-classes, while they faced challenges when acquiring those with verbs falling into other categories. Consequently, it was anticipated that Chinese learners of English at developmental stages would not completely acquire directions of possession transfer in English DOCs involving the syntax-semantics interface. Instead, they might have different degrees of difficulty in processing directions of possession transfer in English DOCs.

### 3.2 Participants

Sixty-three Chinese learners of English from Hunan University and thirty native English speakers (NS) participated in the experiment. To assess the development of L2 proficiency, the L2 learners were categorized into four groups—beginner (BE), post-beginner (PB), intermediate (IN), and advanced (AD)—based on their scores from the Oxford Placement Test (OPT) [36]. Comprising 60 multiple-choice questions with a maximum score of 60, the OPT evaluates

**Table 2. Information about participants.**

|  | BE | PB | IN | AD | NS |
|---|---|---|---|---|---|
| Number of participants | 15 | 17 | 15 | 16 | 30 |
| Mean age<br>(ranges in brackets) | 17.60<br>(16–18) | 17.47<br>(17–18) | 21.20<br>(17–23) | 21.69<br>(20–22) | 27.8<br>(18–45) |
| Duration (years) of learning English | 9.13 | 8.06 | 12.87 | 13.06 | N/A |
| Proficiency test score (ranges in brackets, Max. = 60) | 13.80<br>(10–17) | 21.06<br>(18–28) | 42.47<br>(33–46) | 51.00<br>(48–55) | N/A |

reading, vocabulary, and grammar skills [37]. Detailed information about participants is presented in Table 2.

Approval of this study was granted by the Human Research Ethics Committee of Hunan University. Recruitment took place between March 13, 2023 and November 7, 2023. Informed written consent was obtained from all participants in compliance with the experimental protocols.

The current study used simple linear regressions to assess distinctions between any pair of groups. As depicted in Table 3, all L2 groups exhibit significant differences in their performance on the OPT. Thus, the findings allow for an exploration of within-group development when investigating the L2 acquisition of directions of possession transfer in English DOCs. Fig 1 illustrates the mean and range of scores of each L2 group in the proficiency test.

## 3.3 Instruments and procedures

To address the three research questions, this study utilized comprehension tests to evaluate participants' processing of directions of possession transfer in English DOCs.

In the comprehension tests, participants were tasked with judging the directions of possession transfer in the target constructions. The testing materials comprised 93 target items, of which 21 items (3 types × 7 tokens) are exemplified in Table 4, along with 72 filler items. The format of the twenty-two target items was structured as follows. Seven rightward, leftward, and non-directional verbs were carefully selected based on prior studies [38, 39] and deemed compatible with English DOCs. Each target sentence that contains each verb was presented in the [subject]-[verb]-[the IO]-[the DO] order with animate IOs and inanimate DOs. Following the matrix clause, two additional regions were presented on a region-by-region basis, creating a subordinate clause that continued the prior clause as grammatically as possible. For manipulating directions of possession transfer in the target English DOCs, rightward, leftward, and non-directional verbs were chosen. Note that directions of possession transfer in English

**Table 3. Simple linear regressions of proficiency scores in each parired group.**

|  | β | SE | T-value | P-value |
|---|---|---|---|---|
| BE versus PB | 7.259 | 0.966 | 7.511 | < 0.001* |
| PB versus IN | 21.408 | 0.966 | 22.153 | < 0.001* |
| IN versus AD | 8.533 | 0.980 | 8.704 | < 0.001* |
| BE versus IN | 28.667 | 0.996 | 28.778 | < 0.001* |
| PB versus AD | 29.941 | 0.950 | 31.510 | < 0.001* |
| BE versus AD | 37.200 | 0.980 | 37.942 | < 0.001* |

The left groups were set as reference groups.
*Means p < .05.

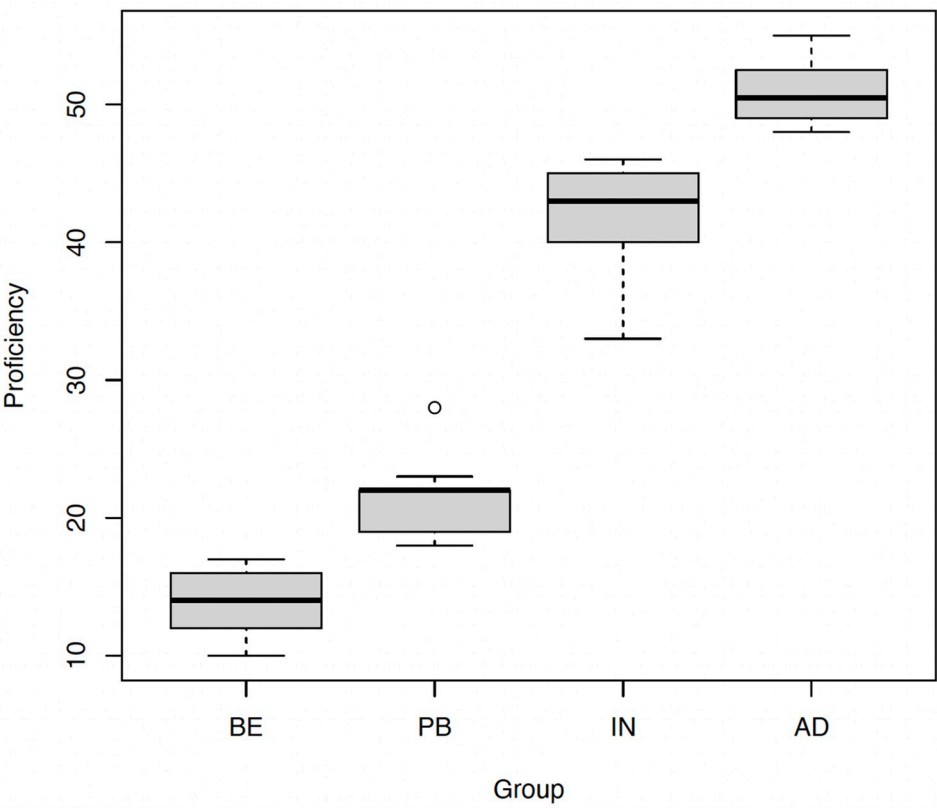

**Fig 1. Proficiency test scores of the four L2 groups.**

DOCs are not determined by the original directions of the verbs. For example, directions of possession transfer in English DOCs with leftward and non-directional verbs remain rightward. To ensure that participants paid attention to the content of the target sentences during the tests, they were required to answer comprehension questions related to the sentences they had just read. The end-of-trial comprehension questions for the target sentences were relevant to directions of possession transfer in English DOCs. For example, the comprehension question of the target sentence 'John posted Mary a letter, which was emotional.' is 'Who finally got the letter?'.

The non-native speaker (NNS) group underwent a comprehension test employing the self-paced reading paradigm, whereas the native speaker (NS) group completed a comprehension test using online questionnaires.

The self-paced reading task was individually administered using a computer-based approach. Prior to the task initiation, participants were instructed to read a vocabulary list and provide the meanings of each word to ensure their familiarity with the task-related vocabulary [20]. The paradigm adopted a segment-by-segment non-cumulative self-paced moving

**Table 4. Example stimuli for the test.**

| Types | Sentences |
|---|---|
| DOCs with rightward verbs | John posted Mary a letter, which was emotional. |
| DOCs with leftward verbs | Sam bought Lucy a rose, which was sweet. |
| DOCs with non-directional verbs | Ben made Kate a chair, which was woody. |

windows task. Participants were presented with each sentence on a computer screen one segment at a time, and they were instructed to read each segment carefully and quickly. To advance to the next segment, participants pressed the space bar on the keyboard after the prior segment disappeared. This process continued until the last segment of the sentence, marked by a full stop, indicating the sentence's end [40]. Upon pressing the space bar after the final segment, a comprehension question appeared on the screen, such as 'Who finally got the letter?', and participants were required to press an appropriate key to indicate their answer. The E-Prime presentation software was used for sentence presentation and data collection. The self-paced reading task was administered in quiet laboratories at two universities in China. Participants received both written and oral instructions before the experiment. Six practice sentences were presented before the main task to familiarize participants with the procedure.

The comprehension task was administered through a web-based survey tool. Each target sentence appeared on a single page, accompanied by a comprehension question concerning the relationship between the IO and the DO, such as 'Who finally got the letter?, similar to the self-paced reading paradigm. Once the task started, participants were restricted from revisiting previous items.

## 3.4 Analysis methods

The analysis focused on the comprehension accuracy for each condition, taking into consideration the factors of group (NS group, NNS group) and verb type (rightward verbs, leftward verbs, non-directional verbs). Logistic mixed-effects models were used via the lmerTest package [41] in R [42]. These models, which are more suitable for analyzing categorical data than ANOVAs [43], incorporated both by-participant and by-item random effects. Likelihood ratio tests were used to determine whether additional parameters significantly improved model fit. Treatment contrasts were established to allow comparisons across different levels of the categorical variables [44], with NNS group and leftward verbs serving as the reference levels for the group and verb type.

## 4 Results

Fig 2 illustrates the comprehension rates for NNS and NS groups across the two experimental sets (mean values and standard deviations are presented in Table 5). All NS participants demonstrated correct comprehension of English DOCs containing three types of verbs, achieving a comprehension rate of approximately 100%. By contrast, NNS participants exhibited a maximum comprehension rate of only 75.7% for English DOCs containing rightward and non-directional verbs, whereas their comprehension rate for those containing leftward verbs fell below 50%.

To examine the influence of group and verb type on participant responses, logistic mixed-effects models were employed to analyze the accuracy rate. The fixed effects included group (NNS, NS), verb type (rightward, leftward, non-directional), and their interactions. Additionally, random effects for both participant and item were considered. In the statistical analysis, a significant main effect of group was observed, indicating higher accuracy rates for native English speakers relative to Chinese learners of English ($\beta = 3.45$, $SE = 0.42$, $z = 8.26$, $p < 0.001$). The main effect of verb type also reached significance, with higher accuracy rates for English DOCs containing rightward verbs ($\beta = 1.48$, $SE = 0.30$, $z = 4.97$, $p < 0.001$) and non-directional verbs ($\beta = 1.09$, $SE = 0.29$, $z = 3.72$, $p < 0.001$) than those with leftward verbs.

Table 6 illustrates the comprehension rate for each group. As shown in Table 4, native English speakers achieved an almost 100% comprehension rate. In contrast, the comprehension rates of Chinese speakers at different L2 proficiency levels increased as L2 proficiency

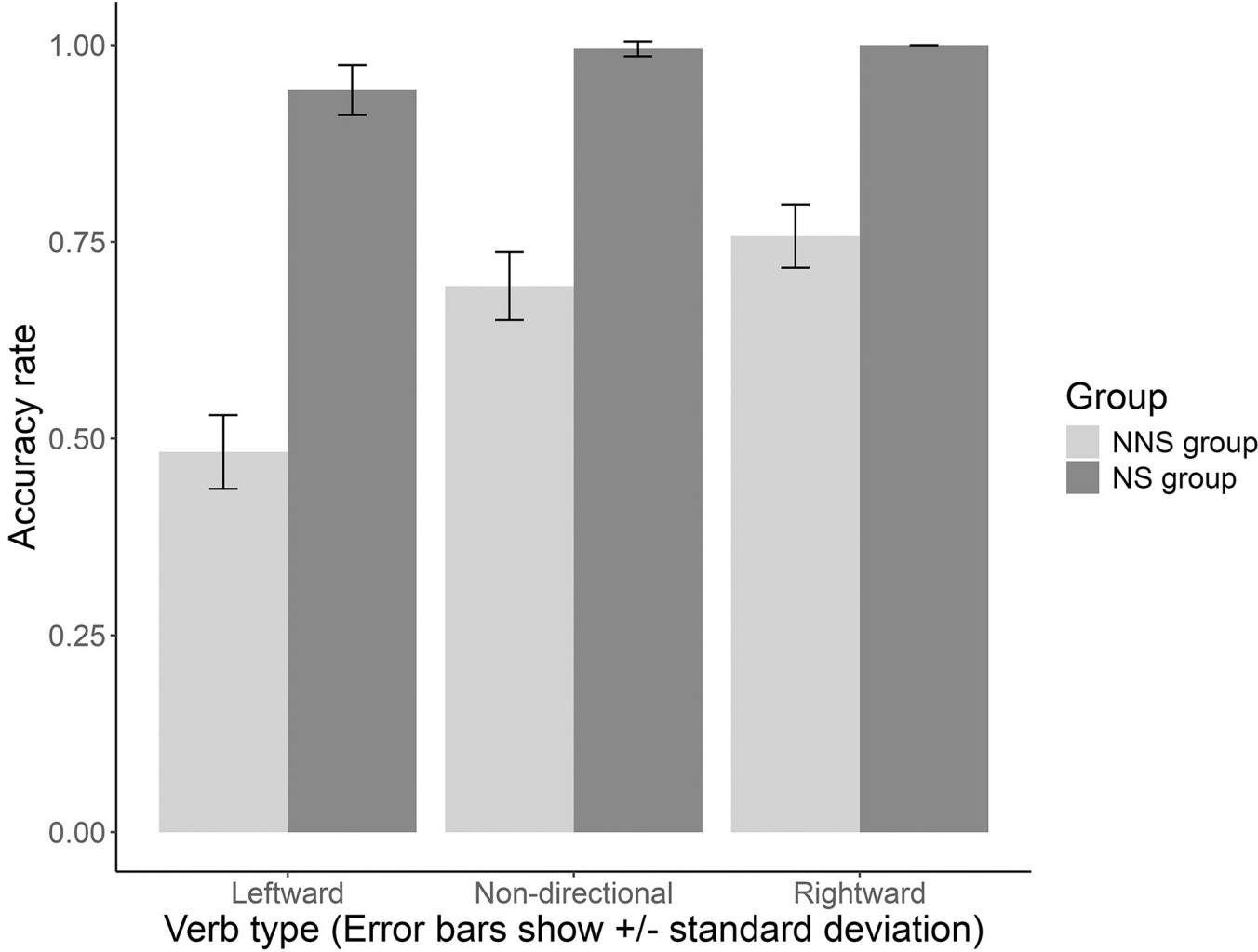

**Fig 2. Accuracy rates for comprehending DOCs in the NNS and NS groups.**

advanced. Specifically, whereas Chinese speakers can be native-like regarding the comprehension rate of directions of possession transfer in English DOCs containing rightward and non-directional verbs, it seems that they permanently deviate from native English speakers regarding those containing leftward verbs. This finding imply that Chinese speakers do not completely acquire directions of possession transfer in English DOCs, and they have difficulty in comprehending directions of possession transfer in English DOCs containing leftward verbs.

We next analyzed data by sub-class of verbs to examine the acquisition of English DOCs with rightward verbs among different groups. Simple linear regression results, presented in Table 7, reveal significant differences between BE and NS and between PB and NS ($p < 0.001$),

**Table 5. Means and standard deviations in comprehending DOCs with respective directions of possession transfer in the NNS and NS groups.**

| Group | Rightward verbs | Leftward verbs | Non-directional verbs |
|---|---|---|---|
| NNS group | 0.757 (0.43) | 0.483 (0.50) | 0.694 (0.46) |
| NS group | 1 (0.00) | 0.943 (0.23) | 0.995 (0.07) |

**Table 6. Means and standard deviations in comprehending DOCs with respective directions of possession transfer in each proficiency group.**

|        | Rightward verbs | Leftward verbs | Non-directional verbs |
|--------|-----------------|----------------|-----------------------|
| BE     | 0.543 (0.50)    | 0.448 (0.50)   | 0.419 (0.50)          |
| PB     | 0.546 (0.50)    | 0.437 (0.50)   | 0.496 (0.50)          |
| IN     | 0.971 (0.17)    | 0.457 (0.50)   | 0.895 (0.31)          |
| AD     | 0.982 (0.13)    | 0.589 (0.49)   | 0.973 (0.16)          |
| NS     | 1 (0.00)        | 0.943 (0.23)   | 0.995 (0.07)          |

but not between IN and NS ($p = 0.797$) or between AD and NS ($p = 0.618$). Regarding learner groups at different proficiency levels, linear regression reveal significant differences between BE and IN, between PB and IN, between BE and AD, and between PB and AD ($p < 0.001$), but not between BE and PB ($p = 0.935$) or between IN and AD ($p = 0.797$). These findings suggest that Chinese speakers begin to acquire English DOCs containing rightward verbs from the intermediate stage of L2 acquisition, although these constructions remain challenging at the beginner and post-beginner stages.

Regarding English DOCs with leftward verbs, as indicated by the simple linear regression results in Table 8, each L2 group significantly differs from native speakers ($p < 0.001$). Regarding groups at different proficiency levels, linear regressions show significant differences between BE and AD ($p = 0.016$), between PB and AD ($p = 0.007$), and between IN and AD ($p = 0.024$), but not between BE and PB ($p = 0.854$), between BE and IN ($p = 0.873$), or between PB and IN ($p = 0.727$). These findings suggest that L2 learners across all proficiency levels face challenges in acquiring English DOCs with leftward verbs, though advanced learners display distinct patterns compared to those at beginner, post-beginner, and intermediate levels.

As for English DOCs containing non-directional verbs, simple linear regression results, as illustrated in Table 9, reveal differences between BE and NS ($p = 2e\text{-}16^*$), between PB and NS ($p = 2e\text{-}16^*$) and between IN and NS ($p = 0.011$), but not between AD and NS ($p = 0.565$). Regarding groups at different proficiency levels, simple linear regression results reveal differences between BE and IN ($p = 2e\text{-}16^*$), between PB and IN ($p = 2e\text{-}16^*$), between BE and AD ($p = 2e\text{-}16^*$) and between PB and AD ($p = 2e\text{-}16^*$), but not between BE and PB ($p = 0.080$) or between IN and AD ($p = 0.080$). Thus, Chinese speakers are capable

**Table 7. Simple linear regressions of DOCs with rightward verbs in each paired group.**

|              | β     | SE    | T-value | P-value    |
|--------------|-------|-------|---------|------------|
| BE versus PB | 0.003 | 0.041 | 0.082   | 0.935      |
| PB versus IN | 0.425 | 0.041 | 10.383  | < 0.001*   |
| IN versus AD | 0.011 | 0.042 | 0.258   | 0.797      |
| BE versus IN | 0.429 | 0.042 | 10.153  | < 0.001*   |
| PB versus AD | 0.436 | 0.040 | 10.826  | < 0.001*   |
| BE versus AD | 0.439 | 0.042 | 10.573  | < 0.001*   |
| BE versus NS | 0.457 | 0.037 | 12.505  | < 0.001*   |
| PB versus NS | 0.454 | 0.035 | 12.931  | < 0.001*   |
| IN versus NS | 0.029 | 0.037 | 0.782   | 0.435      |
| AD versus NS | 0.018 | 0.036 | 0.499   | 0.618      |

The left groups were set as reference groups.

*Means p < .05.

**Table 8. Simple linear regressions of DOCs with leftward verbs in each paired group.**

|  | β | SE | T-value | P-value |
|---|---|---|---|---|
| BE versus PB | -0.011 | 0.058 | -0.185 | 0.854 |
| PB versus IN | 0.020 | 0.058 | 0.350 | 0.727 |
| IN versus AD | 0.132 | 0.058 | 2.26 | 0.024* |
| BE versus IN | 0.010 | 0.059 | 0.160 | 0.873 |
| PB versus AD | 0.152 | 0.057 | 2.687 | 0.007* |
| BE versus AD | 0.142 | 0.058 | 2.423 | 0.016* |
| BE versus NS | 0.495 | 0.051 | 9.625 | < 0.001* |
| PB versus NS | 0.506 | 0.049 | 10.242 | < 0.001* |
| IN versus NS | 0.486 | 0.051 | 9.440 | < 0.001* |
| AD versus NS | 0.354 | 0.050 | 7.019 | < 0.001* |

The left groups were set as reference groups.

*Means $p < .05$.

of mastering English DOCs containing non-directional verbs at the advanced stage of L2 acquisition, whereas they have problems with these constructions when their L2 proficiency is below the advanced level.

## 5 Discussion

The current study delved into the processing of directions of possession transfer in English DOCs by Chinese speakers. The results unveiled two key findings: firstly, Chinese speakers exhibited native-like comprehension, particularly with rightward and non-directional English DOCs, and their proficiency in L2 English positively correlated with this competence. However, they struggled to replicate native-like comprehension with English DOCs containing leftward verbs. Secondly, English DOCs with leftward verbs elicited less accurate responses than those with rightward and non-directional verbs. These outcomes suggest varying levels of difficulty for L2 learners in processing English DOCs containing different verb types, indicative of an indeterminacy in the processing of English DOCs at the syntax-semantics interface by Chinese speakers.

**Table 9. Simple linear regressions of DOCs with non-directional verbs in each paired group.**

|  | β | SE | T-value | P-value |
|---|---|---|---|---|
| BE versus PB | 0.077 | 0.044 | 1.753 | 0.080* |
| PB versus IN | 0.399 | 0.044 | 9.123 | < 0.001* |
| IN versus AD | 0.078 | 0.044 | 1.755 | 0.080* |
| BE versus IN | 0.476 | 0.045 | 10.552 | < 0.001* |
| PB versus AD | 0.477 | 0.043 | 11.090 | < 0.001* |
| BE versus AD | 0.554 | 0.044 | 12.476 | < 0.001* |
| BE versus NS | 0.576 | 0.039 | 14.743 | < 0.001* |
| PB versus NS | 0.499 | 0.038 | 13.312 | < 0.001* |
| IN versus NS | 0.100 | 0.039 | 2.559 | 0.011* |
| AD versus NS | 0.022 | 0.038 | 0.576 | 0.565 |

The left groups were set as reference groups.

*Means $p < .05$.

## 5.1 Comparisons between L1 and L2 speakers of English

The initial inquiry aimed to discern disparities in the processing of directions of possession transfer in English DOCs between native speakers and L2 learners. The results from the present study illuminate significant distinctions between the two groups. Native speakers exhibited accurate comprehension of directions of possession transfer in English DOCs. Conversely, Chinese learners of English did not completely acquire these directions, with English DOCs containing leftward verbs posing more challenges compared to those with rightward and non-directional verbs. Additionally, this study delved into the impact of L2 English proficiency levels. Results indicated that, prior to reaching an intermediate proficiency stage, namely at the initial stage, Chinese learners of English struggled to achieve native-like comprehension of English DOCs containing rightward verbs, while before reaching an advanced stage, similar challenges persisted for those containing non-directional verbs. Across all English proficiency levels, native-like comprehension remained elusive for those containing leftward verbs.

The findings suggest that Chinese learners of English at developmental stage did not attain complete acquisition of English DOCs, manifesting indeterminacy in their processing of such structures. This indeterminacy aligns with the prior study, implying that L2 learners' evaluations of Chinese sentences with wh-EPWs (existential polarity words) involving the syntax-semantics interface also exhibit indeterminacy [45]. The results further accords with the argument that L2 learners' challenges in establishing interface relations in L2 grammars are possibly affected by various variables, including the categorical nature of individual elements involved in the interface relationship, the status of these elements in the target language speaker's grammar, and cross-linguistic influences.

In accordance with the reviewed the Interface Hypothesis [3, 34], it is anticipated that internal interfaces, particularly the syntax-semantics interface, are not problematic and can be eventually acquired by very advanced or near-native speakers. However, the current study presents a nuanced perspective. Whereas intermediate Chinese speakers demonstrated the ability to acquire English DOCs with rightward verbs, and advanced Chinese speakers were proficient in those with non-directional verbs, Chinese speakers at every English proficiency level exhibited difficulties in acquiring English DOCs with leftward verbs. These findings deviate from the expectation that internal interfaces like the syntax-semantics interface are universally acquirable, implying learners' persistent difficulty in establishing English DOCs with leftward verbs in L2 English grammars.

The Interface Hypothesis could offer a plausible explanation for the challenges encountered by L1-Mandarin Chinese L2-English learners in processing English DOCs. According to the Interface Hypothesis, the online integration of syntactic and semantic information demands considerable processing resources, and L2 learners might have limited cognitive resources available [34, 46]. When processing English DOCs, Chinese learners of English might have difficulty in deploying sufficient cognitive resources for integrating verbal event information with constructional meaning. Therefore, they are not able to completely acquire English DOCs. The present study indicates difficulties in the integration of different levels of linguistic knowledge during online processing, implying the indeterminate behavior of different linguistic phenomena within the same interface [47].

The discussion in this subsection answers the first research question regarding the divergence between the L2 grammar and the native grammar. Whereas L2 learners can eventually acquire English DOCs containing rightward and non-directional verbs, they fail to acquire those with leftward verbs, implying that directions of possession transfer in English DOCs remains indeterminate in Chinese speakers' L2 English grammars.

## 5.2 Influencing factors in the processing of double object constructions

Referring to the second research question, L1 influence appears evident in initial L2 English grammars, consistent with the predictions of the Full Transfer Hypothesis [48]. In English DOCs, the direction of possession transfer is exclusively rightward, but in Chinese DOCs, it can be either rightward or leftward. This distinction indicates that while the direction of possession transfer in DOCs aligns with rightward verbs in both languages, it diverges with leftward verbs. For example, in English DOCs (e.g., *John fed Mary a cake*) and Chinese DOCs (e.g., *John wèi le Mary yī-kuài dàngāo*) containing rightward verbs, the direct object (i.e., *cake* or *dàngāo*) is transferred rightward from the subject *John* to the indirect object *Mary*. In contrast, in English DOCs (e.g., *John bought Mary a book*) and Chinese DOCs (e.g., *John mǎi-le Mary yī-ben shū*) containing leftward verbs, the direct object (i.e., *book* or *shū*) is still transferred rightward from the subject to the indirect object in English, while it shifts leftward from the indirect object to the subject in Chinese. Notably, Chinese DOCs containing verbs such as *jiè* (meaning 'lend' or 'borrow') can be either right-directional or left-directional. Our study found that Chinese speakers begin to acquire English DOCs containing rightward verbs at an intermediate stage of L2 acquisition, suggesting that as proficiency increases, they may reach native-like comprehension of the direction of possession transfer shared bewteen L1 and L2. Conversely, Chinese speakers at all L2 proficiency level experience greater difficulties with English DOCs containing leftward verbs, indicating a lack of native-like processing for the direction of possession transfer that differs between their L1 and L2. These findings suggest that when the direction of possession transfer in learners' L1 aligns with that of the L2, native-like L1 processing mechanisms may facilitate L2 processing; however, when there are discrepancies, persistent challenges are likely to arise.

The second factor under consideration is the type of verbs used in DOCs, which can be classified as rightward, leftward and non-directional. According to prior research, rightward verbs denote giving, instantaneous causation of ballistic motion, sending, and future having; leftward verbs imply obtaining; and non-directional verbs are associated with creation [38, 39]. The processing difficulty observed in DOCs containing leftward verbs compared to those containing non-directional and rightward verbs suggests that the semantics of verbs plays a crucial role in the comprehension of English DOCs by Chinese learners of English. This aligns with previous studies [9, 10], highlighting the influence of verb semantics on the L2 acquisition of English DOCs.

L2 proficiency emerges as another important factor influencing the processing of directions of possession transfer in English DOCs. The observed increase in the impact of verb type with proficiency in the L2 group suggests that higher proficiency levels might facilitate L2 predictive processing. This observation resonates with the findings of a self-paced reading study [11], in which the effect of verb-construction integration in the L2 group increases as L2 proficiency advances.

The discussion in this subsection addresses the second research question regarding the influencing factors in the processing of directions of possession transfer in English DOCs by Chinese learners of English. It implies that L1 transfer, verb type and learners' L2 proficiency are three potential factors influencing the processing of English DOCs at the syntax-semantics interface. In essence, the processing of English DOCs with different types of verbs is susceptible to L1 transfer and L2 proficiency, highlighting the multifaceted nature of factors impacting processing beyond the syntax-semantics interface itself.

## 5.3 The indeterminacy of the syntax-semantics interface

In this study, our focus has been on the observation that L2 learners do not fully acquire English DOCs, which is expected given that they are at developmental stages. However, it is

noteworthy that English DOCs containing leftward verbs present more challenges than those with rightward and non-directional verbs. This suggests that the processing of English DOCs at the syntax-semantics interface is not determinate. We propose that verbs in DOCs are indeterminate, encompassing leftward, rightward as well as non-directional variations, leading to the indeterminacy of DOCs. Consequently, Chinese learners of English exhibit indeterminate patterns in the processing of English DOCs. Moreover, we argue that interface constructions, particularly those involving the syntax-semantics interface, are inherently indeterminate. Precisely, the indeterminacy of verbs contributes to the indeterminacy of syntax-semantics interface constructions.

For learners at developmental stages, the indeterminacy within syntax-semantics interface constructions could create challenges in L2 processing. Additionally, factors such as L1 transfer, L2 proficiency and processing resources may collectively impact the L2 processing of the syntax-semantics interface constructions. The explanations are detailed below:

a. L1 Transfer: If the features of a construction in learners' L1 are similar to their L2, the processing mechanisms developed in their L1 may facilitate the processing of the corresponding construction in their L2. Conversely, if the features of their L1 and L2 are different in a construction, the processing of the target construction in their L2 may be susceptible to interference from the construction ingrained in their L1.

b. L2 Proficiency: As learners' proficiency in their target language (L2) increases, it is likely to facilitate more effective processing of the target construction. Higher L2 proficiency enables learners to process the target construction with greater accuracy.

c. Processing Resources: The availability and efficient allocation of cognitive processing resources are crucial during the online processing of the target construction. Limited processing resources may pose challenges for L2 learners, particularly those at developmental stages, influencing their ability to seamlessly integrate and comprehend the target construction.

This account is complementary to the Interface Hypothesis. Whereas the Interface Hypothesis postulates that external interfaces such as the syntax-discourse/pragmatics interface pose greater challenges than internal interfaces such as the syntax-semantics interface, our account places emphasis upon the indeterminacies intrinsic to interfaces themselves. In consistent with White [47], different linguistic phenomena within the same interface might not exhibit uniform behavior. In addition, it is noteworthy that the Interface Hypothesis primarily addresses very advanced stages of L2 acquisition, while our account extends its predictions to encompass L2 learners at developmental stages.

The validity and refinement of our account necessitate further scrutiny in future research. One avenue for exploration is investigating its applicability to other interface constructions. Moreover, it would be intriguing to investigate whether this framework holds true for L2 learners at the very final stages of acquisition and for native speakers. Additional empirical evidence in these aspects will contribute to a more comprehensive understanding of the subtle dynamics involved in language acquisition and processing.

## 6 Conclusions

This study explores the developmental stages of L2 grammar via an examination of Chinese speakers' processing of English DOCs. The findings imply a considerable delay in the L2 acquisition of the syntax-semantics interface, consistent with Sorace's postulation that interfaces pose challenges for L2 learners [1, 2]. Moreover, our findings suggest that the difficulties L2

learners face in the processing of the syntax-semantics interface constructions stem from the indeterminacy within this interface. However, an important limitation lies in the utilization of different paradigms for comprehension tests between L2 learners and native speakers, even though they comprehend the same materials. Future research could address this limitation by comparing online reading times between the two groups in the processing of English DOCs.

## Supporting information

**S1 Dataset.**
(XLSX)

## Author Contributions

**Conceptualization:** Yuxi Li, Tao Zeng, Ze Liu.

**Data curation:** Ze Liu.

**Formal analysis:** Yuxi Li.

**Investigation:** Yuxi Li, Tao Zeng, Ze Liu.

**Methodology:** Yuxi Li, Tao Zeng, Ze Liu.

**Project administration:** Yuxi Li, Tao Zeng.

**Resources:** Tao Zeng.

**Supervision:** Tao Zeng.

**Validation:** Tao Zeng.

**Writing – original draft:** Yuxi Li, Ze Liu.

**Writing – review & editing:** Yuxi Li.

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
