## [Decision Letter · Decision Letter 0]

25 Oct 2024

PONE-D-24-28617Testing the Interface Hypothesis: Evidence from processing directions of possession transfer in double object constructions by L1-Mandarin Chinese L2-English learnersPLOS ONE

Dear Dr. Zeng,

Thank you for submitting your manuscript to PLOS ONE. After careful consideration, we feel that it has merit but does not fully meet PLOS ONE’s publication criteria as it currently stands. Therefore, we invite you to submit a revised version of the manuscript that addresses the points raised during the review process. The study mentions L1 transfer as a possible variable influencing L2 learners' difficulties with DOCs, but this aspect could be explored more thoroughly. Therefore, please provide a more detailed analysis of how L1 Mandarin structures might interfere with or facilitate the acquisition of English DOCs would enhance the explanatory power of the study.

We look forward to receiving your revised manuscript.

Kind regards,

Pushpanathan Thiruvengadam, Ph.D (Academic Editor)

PLOS ONE

Journal Requirements:

1. When submitting your revision, we need you to address these additional requirements. Please ensure that your manuscript meets PLOS ONE's style requirements, including those for file naming. The PLOS ONE style templates can be found at https://journals.plos.org/plosone/s/file?id=wjVg/PLOSOne_formatting_sample_main_body.pdf and https://journals.plos.org/plosone/s/file?id=ba62/PLOSOne_formatting_sample_title_authors_affiliations.pdf 2. We note that your Data Availability Statement is currently as follows: [All relevant data are within the manuscript and its Supporting Information files.] Please confirm at this time whether or not your submission contains all raw data required to replicate the results of your study. Authors must share the “minimal data set” for their submission. PLOS defines the minimal data set to consist of the data required to replicate all study findings reported in the article, as well as related metadata and methods (https://journals.plos.org/plosone/s/data-availability#loc-minimal-data-set-definition). For example, authors should submit the following data: - The values behind the means, standard deviations and other measures reported;- The values used to build graphs;- The points extracted from images for analysis. Authors do not need to submit their entire data set if only a portion of the data was used in the reported study. If your submission does not contain these data, please either upload them as Supporting Information files or deposit them to a stable, public repository and provide us with the relevant URLs, DOIs, or accession numbers. For a list of recommended repositories, please see https://journals.plos.org/plosone/s/recommended-repositories. If there are ethical or legal restrictions on sharing a de-identified data set, please explain them in detail (e.g., data contain potentially sensitive information, data are owned by a third-party organization, etc.) and who has imposed them (e.g., an ethics committee). Please also provide contact information for a data access committee, ethics committee, or other institutional body to which data requests may be sent. If data are owned by a third party, please indicate how others may request data access. 3. Please include your full ethics statement in the ‘Methods’ section of your manuscript file. In your statement, please include the full name of the IRB or ethics committee who approved or waived your study, as well as whether or not you obtained informed written or verbal consent. If consent was waived for your study, please include this information in your statement as well.

Additional Editor Comments:

The study mentions L1 transfer as a possible variable influencing L2 learners' difficulties with DOCs, but this aspect could be explored more thoroughly. Providing a more detailed analysis of how L1 Mandarin structures might interfere with or facilitate the acquisition of English DOCs would enhance the explanatory power of the study.

Reviewers' comments:

Reviewer's Responses to Questions

**Comments to the Author**

1. Is the manuscript technically sound, and do the data support the conclusions?

Reviewer #1: Yes

2. Has the statistical analysis been performed appropriately and rigorously? 

Reviewer #1: Yes

3. Have the authors made all data underlying the findings in their manuscript fully available?

Reviewer #1: Yes

4. Is the manuscript presented in an intelligible fashion and written in standard English?

Reviewer #1: Yes

5. Review Comments to the Author

Reviewer #1: This article provides a well-structured and insightful exploration of the research topic, making a meaningful contribution to the field. The introduction clearly establishes the research objectives, while the findings are both relevant and thoughtfully presented. The methodology is sound, and with a bit more detail on the sampling and data collection processes, it would further strengthen the study’s rigor. Overall, the paper concludes with compelling insights, and expanding on the practical applications and future research potential would leave a lasting impact on the reader.

6. PLOS authors have the option to publish the peer review history of their article (what does this mean?). If published, this will include your full peer review and any attached files.

Reviewer #1: **Yes: **Dr.T. Senthamarai

---

## [Author Response · Author response to Decision Letter 0]

31 Oct 2024

Dear reviewers,

Thank you for your valuable comments and suggestions, which have helped greatly in the improvement of the paper. We have carefully considered each of your comments and have revised the manuscript (highlighted in red), and we also detail below our responses to major points raised in the reviews.

The study mentions L1 transfer as a possible variable influencing L2 learners’ difficulties with DOCs, but this aspect could be explored more thoroughly. Providing a more detailed analysis of how L1 Mandarin structures might interfere with or facilitate the acquisition of English DOCs would enhance the explanatory power of the study.

Response: We acknowledge your valuable point regarding the influence of L1 transfer on the acquisition of English Double Object Constructions (DOCs) by Chinese L2 learners. In response, we have refined the relevant paragraphs in the manuscript, which can be found on pages 41-42 (lines 759-782). Specifically, we elaborated on how Chinese and English DOCs differ in the direction of possession transfer, particularly distinguishing between rightward and leftward verbs. For example, whereas possession transfer is exclusively rightward in English DOCs, Chinese DOCs can feature both rightward and leftward transfer. Our findings suggest that L2 learners can acquire English DOCs containing rightward verbs relatively early, but they struggle with those containing leftward verbs throughout different proficiency levels, likely due to L1 interference.

 By detailing how L1 processing mechanisms either facilitate or interfere with the acquisition of English DOCs depending on the alignment of possession transfer directions, we hope we can address the reviewer’s concern regarding the influence of L1 on L2 acquisition of DOCs.

Best regards,

Tao Zeng

---

## [Editor Report · Decision Letter 1]

4 Nov 2024

Testing the Interface Hypothesis: Evidence from processing directions of possession transfer in double object constructions by L1-Mandarin Chinese L2-English learners

PONE-D-24-28617R1

Dear Dr. Zeng,

We’re pleased to inform you that your manuscript has been judged scientifically suitable for publication and will be formally accepted for publication once it meets all outstanding technical requirements.

Kind regards,

Pushpanathan Thiruvengadam, Ph.D

Academic Editor

PLOS ONE

Additional Editor Comments (optional):

No comments
---

## [Editor Report · Acceptance letter]

21 Nov 2024

PONE-D-24-28617R1 

PLOS ONE

Dear Dr. Zeng, 

I'm pleased to inform you that your manuscript has been deemed suitable for publication in PLOS ONE. Congratulations! Your manuscript is now being handed over to our production team.

Kind regards, 

on behalf of

Dr. Pushpanathan Thiruvengadam 

Academic Editor

PLOS ONE